# Multi-Task Learning of Query Generation and Classification for Generative Conversational Question Rewriting

**Sarawoot Kongyoung[1], Craig Macdonald[2], Iadh Ounis[2]**
University of Glasgow, UK
[1]s.kongyoung.1@research.gla.ac.uk
[2]{craig.macdonald,iadh.ounis}@glasgow.ac.uk

## Abstract

In conversational search settings, users ask questions and receive answers as part of a conversation. The ambiguity in the questions is a common challenge, which can be effectively addressed by leveraging contextual information from the conversation history. In this context, determining topic continuity and reformulating questions into well-defined queries are crucial tasks. Previous approaches have typically addressed these tasks either as a classification task in the case of topic continuity or as a text generation task for question reformulation. However, no prior work has combined both tasks to effectively identify ambiguous questions as part of a conversation. In this paper, we propose a Multi-Task Learning (MTL) approach that uses a text generation model for both question rewriting and classification. Our models, based on BART and T5, are trained to rewrite conversational questions and identify follow-up questions simultaneously. We evaluate our approach on multiple test sets and demonstrate that it outperforms single-task learning baselines on the three LIF test sets, with statistically significant improvements ranging from +3.5% to +10.5% in terms of F1 and Micro-F1 scores. We also show that our approach outperforms single-task question rewriting models in passage retrieval on a large OR-QuAC test set.

## 1 Introduction

Conversational Question Answering (QA), which simulates human dialogues in information-seeking tasks, necessitates resolving ambiguities in the user queries based on the conversation history (Kundu et al., 2020). Figure 1 exemplifies the possible ambiguities that might arise from a typical dialogue with a user, highlighting the importance of addressing these ambiguities based on the conversation history. For example, in the question $q_3$, "Was he the owner of the paper?", the "he" and "the paper" referents are context-dependent, typically clarified by prior questions ($q_1$, $q_2$) and answers ($a_1$, $a_2$).

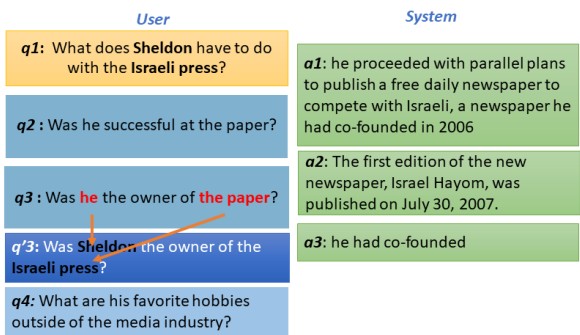

Figure 1: An illustrative example of dialogue.

To address such ambiguities, a number of approaches have been explored, such as follow-up question identification (Bertomeu et al., 2006; Kirschner and Bernardi, 2007, 2009; Kundu et al., 2020) and conversational question rewriting (Lin et al., 2020; Mele et al., 2021; Ren et al., 2018; Vakulenko et al., 2021a,b; Voskarides et al., 2020; Yu et al., 2020). Indeed, the follow-up question identification approaches have been used in Conversational QA to enhance the interpretation of the user's intent and context. The main objective of these approaches is to determine whether a follow-up question is linked to the previous conversation history or not. If it is determined that the follow-up question is related to the previous conversation, the system can leverage the context provided by the conversation to generate more precise and relevant responses. In Figure 1, a follow-up question ($q_3$) is classified as *valid* if it can be linked to the previous conversation ($q_1, a_1, q_2, a_2$), else it is classified as *invalid* (e.g., $q_4$). For example, the three-way attentive pooling network approach (Kundu et al., 2020) predicts this continuity by analysing the candidate question and conversation history, outperforming other models such as BiLSTM, CNN, and BERT (Devlin et al., 2019). In this paper, we use the three-way attentive pooling network as our strongest baseline.

On the other hand, conversational question rewriting approaches have been employed in

Conversational QA to enhance the accuracy of the retrieved information by reformulating the user's original question. These approaches usually involve modifying the original question or generating new queries that better represent the user's intent by transforming a concise conversational question into a fully-grown, contextualised ad-hoc query. For example, in Figure 1, the question $q_3'$ "Was Sheldon the owner of the Israeli press?" is a rewrite of $q_3$ "Was he the owner of the paper?", based on the conversation history $(q_1, a_1, q_2, a_2)$. A T5 (Raffel et al., 2020) model can be fine-tuned to automatically reformulate the question by injecting information that exists in the context into a fully defined query (Lin et al., 2020). Previous work, such as (Lin et al., 2020; Mele et al., 2021; Ren et al., 2018; Vakulenko et al., 2021a,b; Voskarides et al., 2020; Yu et al., 2020), focused on conversational question rewriting; however, they did not address the task of follow-up question identification. In this paper, to alleviate ambiguities in conversational QA, we investigate the combination of both the follow-up question identification task and the conversational question rewriting task into a single framework, thereby also improving the effectiveness of both tasks.

Recently, Multi-Task Learning (MTL) has emerged as an effective approach for simultaneously learning numerous related tasks (Ide and Kawahara, 2021; Kongyoung et al., 2022). MTL can be used to increase a system's performance on the text generation task by leveraging classification tasks. For example, Ide and Kawahara (2021) adopted an MTL approach using a text generation BART (Lewis et al., 2019) model, which jointly learns a classification task and a text generation task by sharing the learner, and showed an improvement in an emotion-aware dialogue response generation model. Text generation models like T5 have also been effectively used for multi-task learning involving both text generation and classification tasks, such as passage ranking/re-ranking and answer generation/extraction (Kongyoung et al., 2022; Lee et al., 2022; Jiang et al., 2022). Given their effectiveness on various tasks, we argue that text generation models – BART and T5 – can be tailored for the follow-up question identification and conversational question rewriting tasks.

Our intuition is that by combining follow-up question identification and conversational question rewriting, the system's response accuracy and

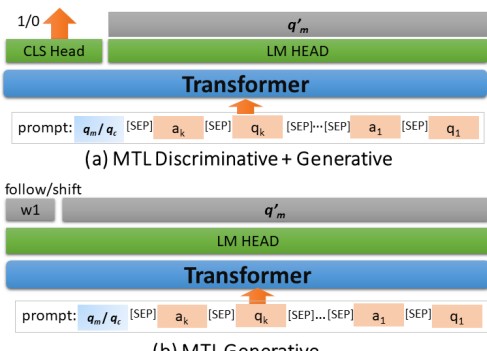

Figure 2: Comparison of MTL models: (a) a discriminative+generative model with separate heads for the classification and question rewriting tasks, and (b) a generative model with a combined token generation for both tasks.

relevance can be enhanced. Indeed, by identifying connections between the user's questions, addressing ambiguities, and leveraging the conversation's context, the system can refine its understanding of the user intent and can provide more precise and relevant responses. Our text generation models leverage the Multi-Task Learning of the conversational question rewriting and classification tasks to identify whether a question is a follow-up to the previous question and, accordingly, reformulate a question using the dialogue context. To the best of our knowledge, no prior work has inherently combined both tasks to more effectively address ambiguity in conversational questions.

Our contributions are as follows: (1) We leverage Multi-Task Learning with a text generation model to effectively address the tasks of follow-up question identification and conversational question rewriting; (2) Using the recent LIF dataset (Kundu et al., 2020), we compare our models to two recent baselines from the literature, and show that our Multi-Task Learning BART model yields the best F1 and Macro-F1 performance improvements over the strongest baseline, three-way attentive pooling, with statistically significant improvements ranging from 3.5% to 10.5% on all LIF test sets; (3) Our proposed Multi-Task Learning T5 model significantly outperforms the single-task learning of question rewriting models for passage retrieval on the OR-QuAC test set.

## 2 MTL: Classification & Generation

We define the follow-up question identification and conversational question rewriting tasks in Sections 2.1 & 2.2. An overview of the proposed text generation model follows in Section 2.3.

## 2.1 Follow-up Question Identification

Following Kundu et al. (2020), we consider the following inputs: a list of previous questions, a list of their corresponding ground truth answer (respectively denoted as $[q_1, q_2, \ldots q_k]$ and $[a_1, a_2, \ldots a_k]$ where $k$ is the number of the previous question-answer pair in the conversation history), and a candidate follow-up question $q_c$. Given these inputs, the first task we address is to predict whether or not the candidate follow-up question $q_c$ is a valid follow-up question. Figure 1 exemplifies the follow-up question identification task, showing a history of length $k = 2$ with the corresponding questions and answers. In particular, as this task is a binary classification task, the aim of a follow-up question identification approach is to classify a question $q_c$ as a *valid* follow-up question or as *invalid*.

## 2.2 Conversational Question Rewriting

Following Elgohary et al. (2019), given a conversation history $H_k$ consisting of a list of $k$ questions and a list of ground truth answer pairs, i.e $H_k = [\langle q, a \rangle]$, the task is to generate a rewrite $q'_m$ for the next question $q_m$ based on $H_k$. Because $q_m$ is part of the conversation, its meaning frequently includes references to parts of $H_k$. A valid $q'_m$ should be self-contained: i.e. a correct answer to $q'_m$ without the history $H_k$ is a correct answer to $q_m$ with the history $H_k$. Figure 1 exemplifies the conversational question rewriting task, showing a history of length $k = 2$ with the corresponding questions and answers. The question $q_3$ omits the first question (replacing the pronoun "he" with Sheldon and replacing "the page" with Israeli press ). Hence, to address this task, the system needs to resolve any omission by using history $H_k$. Next, we describe the MTL approaches to combine the follow-up question identification with the conversational question rewriting.

## 2.3 Model Overview

To tackle the tasks described in Sections 2.1 and 2.2, we propose classification and question rewriting models that leverage historical questions to identify whether a candidate question $q_c$ is a follow-up to the previous question and to reformulate the current question $q_m$. Our proposed method uses models including BART (Lewis et al., 2019) and T5 (Raffel et al., 2020), which are large pre-trained language models designed for text generation. Such text generation approaches can be trained to generate a meaningful textual response based on some input text. Moreover, like BERT, the pre-trained BART and T5 models can be fine-tuned to perform a variety of downstream tasks.

In addition, the manner in which a text generation model is used in classification tasks can differ, as they can be fine-tuned as discriminative or generative models. In a discriminative setup, the model is adapted for binary classification by adding a fully-connected layer with two output neurons (corresponding to each class) upon a special [CLS] token in BERT, or the last token in BART. In contrast, a generative setup reframes NLP tasks as text generation tasks - for instance, classification is performed by examining what text is generated and the corresponding likelihood.

**An MTL Text Generation Approach:** To adapt an MTL approach to a text generation model for jointly learning from both the classification and question rewriting tasks, the model can be used in either a discriminative+generative or in a generative setup as shown in Figure 2. A discriminative+generative MTL model makes predictions by applying a CLS head to create a score for a classification task and an LM head to generate the tokens for a question rewriting task (Ide and Kawahara, 2021) as shown in Figure 2 (a). In contrast, a generative MTL model makes predictions by generating the first token for a classification task and the follow-up tokens for a query rewriting task as shown in Figure 2 (b). Notably, while text generation models like BART can function in either the discriminative+generative or generative MTL setups, the T5 model is exclusively applicable as a generative MTL model (Raffel et al., 2020).

In particular, when fine-tuning the T5 model for a downstream task, a *prefix text* is required – for example "translate English to German:" might be used for a translation task. Indeed, the text generation models have been shown to achieve state-of-the-art performances in classification (Lewis et al., 2019; Raffel et al., 2020), as well as in document re-ranking – by ranking based on the likelihood of generating a particular token (Nogueira et al., 2020; dos Santos et al., 2020) (outperforming BERT models) and even in arithmetic tasks (Nogueira et al., 2021). Hence, for the MTL of both the follow-up question identification and question rewriting tasks we choose the MTL generative version of the BART

and T5 models. However, for comparison purposes, we also deploy the discriminative+generative versions of the BART models in our experiments.

More precisely, we deploy generative MTL models to capture the relation between the question $q_c/q_m$ and the contextual information in the conversation history, including the historical question(s) $\{q_1, q_2, \ldots q_k\}$, and the historical answer(s) $\{a_1, a_2, \ldots a_k\}$, as shown in Figure 2 (b). In particular, let $Gen(\cdot)$ denotes a generative transformation function by taking the input sequence as follows:

$$Gen(\text{``prompt} : \text{''} q_c \text{``[SEP]''} H_k) \quad (1)$$

$$Gen(\cdot) = w_1, w_2, ..., w_n \quad (2)$$

where "prompt:" is a prefix text, and "[SEP]" is a special token. The model is then fine-tuned to generate the target tokens length $n$ as shown in Equation (2), for the token $w_1$ namely "follow" or "shift"[1] depending on whether the candidate question is a valid follow-up to the previous question or not, while the follow-up tokens $w_2, ..., w_n$ are the output sequence for the target query. In particular, a prompt function, as shown in Equation (1), is employed to format and combine the question ($q_c$) and the conversational history ($q_1...q_k, a_1...a_k$), creating a well-structured input sequence for $Gen(\cdot)$. Subsequently, $Gen(\cdot)$ generates the contextual representation $h$. Once the contextual representation $h$ is obtained, it is used by the $Gen(\cdot)$ decoder. This decoder takes the previously generated tokens as input and performs attention over $h$, enabling it to generate the subsequent token. Specifically, when given the tuple $\langle q_c, q_1...q_k, a_1...a_k \rangle$, our training objective aims to minimise the following loss function:

$$\mathcal{L}_{gen} = \sum_{i=0}^{M} \log P(w_i|h, w_{:i}) \quad (3)$$

where $M$ is the number of tokens in the target sequence, which consists of the ground-truth follow-up question identification token $w_1$ followed by the tokens in the manually rewritten question $q'_m$. In addition, $q'_{m_i}$ refers to the $i^{th}$ token in $q'_m$ when $i \geq 2$ and the token $w_0$ is the beginning of the sequence token (). Our goal is to train the model to generate subsequent tokens using the ground-truth follow-up question identification

---

[1] We choose "follow" and "shift" as target tokens because T5 tokenises sequences using the SentencePiece approach (Kudo and Richardson, 2018), which splits the word "invalid" into two subwords.

| | LIF + CANARD | | | LIF | | OR-QuAC |
|---|---|---|---|---|---|---|
| | Train | Dev | Test-I | Test-II | Test-III | Test |
| #questions | 62,839 | 4914 | 5,992 | 5,247 | 2,685 | 5571 |
| #valid follow-up | 22,056 | 1,559 | 1,940 | 1,940 | 1,940 | - |
| #invalid follow-up | 40,783 | 3,355 | 4,052 | 3,307 | 745 | - |

Table 1: Statistics of the used datasets

and the rewritten question. The loss function $\mathcal{L}_{gen}$ measures the discrepancy between the generated tokens and the ground-truth tokens. To summarise, our MTL method involves fine-tuning the generative model to simultaneously generate sequence tokens for the follow-up question identification and conversational question rewriting.

In addition to the implementation of the generative MTL model, we also incorporate a discriminative+generative MTL model as shown in Figure 2 (a), and described in Appendix A.1.

## 3 Experimental Setup

We list our research questions in Section 3.1, introduce the used datasets in Section 3.2 and present our baselines in Section 3.3

### 3.1 Research Questions

We address two main research questions:
**RQ1:** Does the MTL approach of question rewriting and classification using text generation models outperform the single-task learning (STL) of text generation models and existing baselines for the follow-up question identification task?
**RQ2:** Does the MTL approach of question rewriting and classification using text generation models outperform the single-task learning (STL) of text generation models in the context of the conversational question rewriting and passage retrieval tasks?

### 3.2 Datasets

We experiment using the LIF dataset (Learning to Identify Follow-Up) (Kundu et al., 2020) and the CANARD dataset (Context Abstraction: Necessary Additional Rewritten Discourse) (Elgohary et al., 2019), which are a recent adaptation of the well-known QuAC Conversational QA dataset (Choi et al., 2018). For training the models to address both tasks, in the training and development sets, we integrate LIF and CANARD by picking only the candidate questions from the LIF dataset that exist in the CANARD dataset. To evaluate the models in the follow-up question identification task, we use the three test sets of the LIF dataset, namely Test-I, Test-II, and Test-III. In all three test sets,

the valid follow-up questions (label = 1) are constructed from the "should ask follow-up question" instances in the QuAC dataset. For the invalid follow-up questions (label = 0), Test-I combines the invalid instances from Test-II & Test-III. For Test-II, questions with a high similarity to the current passage are sampled from other conversations. On the other hand, for Test-III, the invalid follow-up questions are sampled from the non-follow-up questions of the same conversation in QuAC. For the question rewriting task, we use the test sets of the OR-QuAC dataset. We also aggregate a passage collection from the OR-QuAC dataset (Qu et al., 2020), which is an aggregation of three existing datasets consisting of QuAC, CANARD, and Wikipedia, to evaluate the passage retrieval performance of the conversational question rewriting task. This allows us to assess our model's performance across both the conversational question answering and passage retrieval tasks. For further information about the used datasets, we also provide a summary of their statistics in Table 1.

### 3.3 Baselines and Implementation

**Follow-up question identification task:** We only include neural models as our baselines since the existing rule-based and statistical machine learning models have been shown to be much less effective in the follow-up question identification task in a previous study (Kundu et al., 2020). Indeed, as baselines, we choose the strongest baseline in the previous study (Kundu et al., 2020), *BERT*, as well as the state-of-the-art (SOTA) *three-way attentive pooling* model from the same study. For the three-way attentive pooling model, we reproduce the model and its evaluation results provided by Kundu et al. (2020). We additionally compare our MTL of the generative BART and T5 models with the *Single-Task Learning (STL) of T5 and BART*. The STL models are only learned to predict whether a given question is a follow-up question. We additionally compare our generative BART model with the *discriminative+generative* version of BART, as described in Section 2.3.

**Conversational question rewriting task:** We compare our query rewriting methods with the following models: *Raw:* The user's original current question; *Manual:* The questions are written by humans from the CANARD dataset. We also compare our proposed methods with the *Single-Task Learning (STL) of BART and T5*, which are learned to

only generate the rewritten question $q'_m$. We compare them in terms of both conversational question rewriting and passage retrieval effectiveness.

**Hyperparameter settings and evaluation metrics:** We provide a description of the hyperparameter settings and the used evaluation metrics in Appendix A.2 and A.3, respectively.

## 4 Experimental Results

We now address RQs 1 & 2, and conclude with a qualitative analysis.

### 4.1 RQ1: Follow-up Identification

We investigate the performance of the baselines described in Section 3.3 in comparison to our proposed MTL text generation models for follow-up question identification on all three test sets of the LIF dataset. Table 2 compares our proposed MTL text generation models to the baselines on all three test sets of the LIF dataset.

**Comparison of the MTL Models with the Baselines for Follow-up Question Identification:** From the table, on three test sets of the LIF dataset, we see that the generative MTL BART classifier model achieves the highest recall, F1, and Macro-F1, except MTL T5 in Test-III for Macro-F1. While the best precision scores on all three test sets are obtained by the three-way attentive pooling, discriminative+generative MTL BART, and MTL T5, respectively. Within the table, on all three test sets, our proposed generative MTL BART, significantly outperforms the baseline, BERT, three-way attentive pooling, STL BART (both discriminative and generative), and STL T5 in terms of F1 and Macro F1, except MTL T5 in Test-III for Macro-F1. These results suggest that the generative MTL BART classifier model exhibits strong overall performance and exceeds other models in terms of recall, F1 score, and Macro-F1 score across most test sets. This highlights its ability to accurately predict true positive instances (*valid* follow-up question) while maintaining a good balance between precision and recall.

**Comparison of the MTL Models with STL:** We observe that the generative MTL BART classifier model significantly outperforms the STL BART (both discriminative and generative) models in terms of F1 and Macro-F1 on all three test sets. This indicates that the MTL approach, which jointly trains the model on the follow-up question identification and conversational question

| | Models | Test-I | | | | Test-II | | | | Test-III | | | |
|---|---|---|---|---|---|---|---|---|---|---|---|---|---|
| | | P | R | F1 | Macro-F1 | P | R | F1 | Macro-F1 | P | R | F1 | Macro-F1 |
| STL | BERT | 70.7 | 79.5 | 74.9†‡ | 80.8†‡ | 85.6 | 79.5 | 82.5† | 86.4† | 80.2 | 79.5 | 79.9† | 64.2†‡ |
| | 3-way AP | **71.6** | 70.0 | 71.6†‡ | 79.2†‡ | 74.4 | 76.8 | 75.6†‡ | 80.5†‡ | 79.7 | 70.0 | 74.6†‡ | 60.4†‡ |
| | BART(dis) | 69.7 | 79.4 | 74.2†‡ | 80.3†‡ | 85.7 | 79.4 | 82.5†‡ | 86.4† | 78.9 | 79.4 | 79.1† | 62.1†‡ |
| | BART(gen) | 71.0 | 79.7 | 75.1†‡ | 81.0† | 87.4 | 79.7 | 83.4† | 87.1† | 79.1 | 79.7 | 79.4† | 62.5†‡ |
| | T5 | 69.9 | 83.0 | 75.9† | 81.4† | 85.3 | 83.0 | 84.2† | 87.5† | 79.5 | 83.0 | 81.2† | 64.0† |
| MTL | BART (dis+gen) | 71.5 | 84.6 | 77.5 | 82.6 | **86.3** | 84.6 | 85.4 | 88.5 | 80.6 | 84.6 | 82.5 | 66.4 |
| | BART (gen) | 70.3 | **87.3** | **77.9** | **82.7** | 84.9 | **87.3** | **86.1** | **88.9** | 80.4 | **87.3** | **83.7** | 66.9 |
| | T5 | 71.3 | 83.5 | 76.9† | 82.2 | 85.4 | 81.6 | 83.5† | 87.1† | **84.6** | 77.6 | 80.9† | **68.9** |

Table 2: Results for follow-up question identification. † denotes a performance that is significantly worse than the MTL BART model (McNemar's test, $p < 0.05$); ‡ denotes a performance that is significantly worse than the MTL T5 model (McNemar's test, $p < 0.05$). 3-way AP denotes the Three-Way Attentive Pooling. The highest value for each measure is highlighted.

rewriting tasks, provides a notable advantage over the STL approach for the BART classifier model. However, the MTL T5 model does not outperform the STL T5 model in terms of F1 and Macro-F1, but the two models are not significantly different.

**Comparison Between Our Generative MTL Models:** We observe that the MTL BART classifier model significantly outperforms the MTL T5 model in terms of F1 on all three test sets, and also significantly outperforms MTL T5 on Macro-F1 score on Test-II. Comparing the discriminative+generative and generative MTL BART models, we find that there is little difference between the effectiveness of the two versions of the MLT BART models on both F1 and Macro-F1 scores on all three test sets.

Therefore, in response to RQ1, we find that our MTL text generation model, BART, jointly learned through both the follow-up question identification and conversational question rewriting tasks has the best overall effectiveness, yielding statistically significant improvements in terms of F1 and Macro-F1 over the baselines, on each of the three test sets of the LIF dataset.

### 4.2 RQ2: Question Rewriting

Next, we examine the effectiveness of the conversational question rewriting models including our proposed MTL text generation models, and those listed in Section 3.3 (STL BART, STL T5, and discriminative+generative MTL BART) on the test set of the OR-QuAC dataset. Table 3 presents the effectiveness of various question reformulation models for conversational question rewriting, evaluated based on the ROUGE-1 recall and BLEU scores. Furthermore, the models' effectiveness for passage retrieval is evaluated when applied with the

BM25 [2] retrieval model. The effectiveness of the monoT5 re-ranker for the same 1000 retrieved passages is listed in the same row. The effectiveness of the manually rewritten questions can be seen as an upper bound for the question rewriting methods.

**Comparison of the MTL Models with the Baselines for Conversational Question Rewriting:** In Table 3, we observe that the MTL T5 model achieves the highest ROUGE-1 score by significantly outperforming all STL baselines, demonstrating its superior performance in capturing the recall of the rewritten questions at the unigram level (individual words). On the other hand, the generative MTL BART model achieves the highest BLEU score by significantly outperforming all STL baselines, indicating its effectiveness in measuring the similarity between the generated texts and the reference texts (the manually rewritten questions). Moreover, we can observe that the generative MTL BART model outperforms the STL BART model in terms of both the ROUGE and BLEU scores. This indicates that the generative MTL BART model achieves a better performance in the conversational question rewriting task compared to the STL BART model. Comparing the MTL T5 model to the STL T5 model, the MTL T5 model achieves a higher ROUGE-1 score, indicating a better performance in conversational question rewriting. However, both models have the same BLEU score.

**Comparison of the MTL Models with Baselines for Passage Retrieval:** From Table 3, we observe that our MTL T5 generative model achieves the highest MAP, MRR, and Recall@1000, and does significantly improve over the MTL BART model and all the STL models in both first stage retrieval and re-ranking. Comparing the MTL and

[2] We also conducted experiments with the DPH retrieval model, which yielded similar trends.

| | Models | Question Rewriting | | First Stage (BM25) | | | | Re-ranker (monoT5) | | |
|---|---|---|---|---|---|---|---|---|---|---|
| | | ROUGE-1 | BLEU | MAP | MRR | R@1000 | NDCG | MAP | MRR | NDCG |
| Raw | | 62.82†‡ | 36.01†‡ | 0.0410†‡ | 0.0424†‡ | 0.2335†‡ | 0.0733†‡ | 0.0786†‡ | 0.0809†‡ | 0.1059†‡ |
| STL | BART | 72.91†‡ | 48.15† | 0.1517†‡ | 0.1617†‡ | 0.5576†‡ | 0.2257†‡ | 0.2438†‡ | 0.2580†‡ | 0.3046†‡ |
| | T5 | 73.22‡ | 45.86† | 0.1720‡ | 0.1843‡ | 0.6055‡ | 0.2524‡ | 0.2783‡ | 0.2957‡ | 0.3430‡ |
| MTL | BART(dis+gen) | 73.12‡ | 48.00† | 0.1571‡ | 0.1685‡ | 0.5790‡ | 0.2348‡ | 0.2562‡ | 0.2708‡ | 0.3189‡ |
| | BART(gen) | 73.56‡ | **48.79** | 0.1646‡ | 0.1760‡ | 0.6006‡ | 0.2440‡ | 0.2661 ‡ | 0.2823‡ | 0.3309‡ |
| | T5 | **74.12** | 45.86† | **0.2008** | **0.2150** | **0.6373** | **0.2829** | **0.3106** | **0.3302** | **0.3764** |
| Manual | | 100.00 | 100.00 | 0.2486 | 0.2682 | 0.8012 | 0.3540 | 0.3811 | 0.4066 | 3295.0 |

Table 3: Comparison between the MTL models and the query rewriting baselines. † denotes a performance that is significantly worse than the MTL BART model (paired t-test, $p < 0.05$); ‡ denotes a performance that is significantly worse than the MTL T5 model (paired t-test, $p < 0.05$).

STL models, we observe that both the MTL T5 and BART models significantly outperform their corresponding STL models. Contrasting the performances of the discriminative+generative with those of the generative MTL BART models, we find that there is little difference between the effectiveness of the two versions of the MLT BART model.

**Comparison of our Proposed MTL Models with the Baselines for both tasks:** Our MTL T5 model demonstrates its effectiveness in both tasks. It not only captures the recall of the rewritten questions at the unigram level but also enhances passage ranking, resulting in our MTL T5 model outperforming both the MTL BART model and all of the STL models, yielding statistically significant improvements on both tasks. To illustrate these findings, we provide a further qualitative analysis in Section 4.3.

In answer to RQ2, we conclude that applying the Multi-Task Learning of question rewriting and classification to the T5 model improves the passage retrieval performance, yielding statistically significant improvements over the MTL BART model and all the STL models.

Q1: How did the Disco Volante begin?
A1: Due to artwork delays and the band members' many side-projects, it was four years
before Disco Volante was released, in October 1995.
Q2: How did they perform commercially?
A2: Not all critics were impressed with the album, with The Washington Post describing it as "an **album of cheesy synthesizers**."
Q3: What followed this album?
Manual (Q'3): What followed the **Disco Volante** album?
MTL T5: shift What followed the Disco Volante album?
STL T5: What followed the album of **Cheesy Synthsynergics**?

Figure 3: Comparison of question rewriting models.

### 4.3 Analysis

In this section, we conduct a qualitative analysis to bolster support for our findings concerning the performance of our MTL T5 model in comparison to the STL T5 model, as discussed in Section 4.2. The purpose of this qualitative analysis is to further validate our results and to shed additional lights on the advantages of the MTL approach in the follow-up question identification and conversational question rewriting tasks. First, we present an example of dialogue that exemplifies the distinct advantages derived from the use of the MTL approach in our model, specifically in conversational question rewriting. Next, we proceed to compare the differences in NDCG scores between our proposed MTL model and the STL model for passage retrieval.

**Conversational Question Rewriting:** To illustrate the advantages of the MTL approach in the follow-up question identification and conversational question rewriting tasks, we provide an example dialogue carefully selected based on the highest ROUGE-1 score achieved by our MTL T5 model. This example dialogue, shown in Figure 3, clearly showcases the benefits of employing the MTL strategy.

This example consists of a conversation history with two turns (Q1, A1, Q2, A2), the current question (Q3), the manually rewritten question (Q'3), the rewritten question of our MTL T5 model, and the rewritten question of the STL T5 model. The MTL T5 model successfully predicted the word "shift" indicating an invalid follow-up question, as Q3 deviates from the previous conversation's topic of album performance and instead inquires about subsequent events. The MTL T5 model exhibits a superior performance in predicting invalid follow-up questions, demonstrated by the model achieving the best Macro-F1 score on the Test-III LIF dataset (as described in Section 4.1). Notably, this test

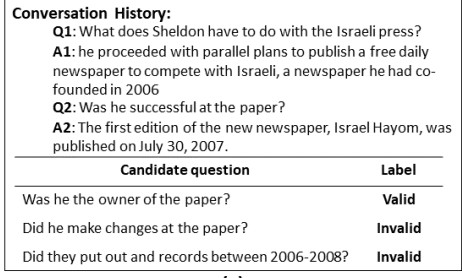

**(a)**

**(b)**

Figure 4: Examples of dialogue differences in terms of NDCG for queries in the OR-QuAC query set. (a) a higher NDCG for MTL T5 wrt STL T5 (b) a higher NDCG for STL T5 wrt MTL T5.

set comprises sampled invalid follow-up questions from the *same conversation*, making this achievement particularly noteworthy. This prediction helps the model to differentiate and choose the accurate entity "Disco Volante" instead of the misleading prediction "Cheesy Synthsynergics" made by the STL T5 model. This demonstrates the ability of the MTL T5 model to better leverage and interpret the context of the conversation, leading to more accurate predictions and an improved performance.

**Passage Retrieval:** We also compare the differences in terms of NDCG scores when using a BM25 ranking model with our proposed MTL model in comparison to using it with the STL model in passage retrieval (MTL T5 vs. STL T5). Figure 4 shows two examples of dialogues selected using the difference in NDCG scores between MTL T5 and STL T5 on the OR-QuAC query set. Overall, MTL T5 outperforms STL T5 for 781 questions, while the opposite was true for 497 questions. Following Macdonald et al. (2021), we only consider differences larger than 0.15 absolute NGCG when inspecting the effect of the MTL approach. Hence, this analysis demonstrates that our proposed MTL T5 model exhibits a superior performance over the STL T5 model in passage retrieval. To further illustrate this point, we closely examine the predictions made by the MTL T5 model in Figure 4 (a). It is clear that the MTL T5 model successfully identi-

fies a candidate question as a valid follow-up to the previous question, thereby demonstrating its capability to potentially aid in the correct resolution of the omitted entity (Odissi). On the other hand, in Figure 4 (b), the candidate question "where did he go to school" would not have logically followed the previous question "did he have siblings". However, the MTL T5 model predicted this candidate question as a valid follow-up question; hence this could lead the model to incorrectly resolve the entity (Roy Acuff). As a result, we can observe that the effectiveness of the follow-up question identification task does influence the conversational question rewriting task performance.

## 5 Related Work

We discuss related work in Conversational QA.

**Conversational QA:** Conversational QA, which involves the correct interpretation of questions within an ongoing dialogue, has been addressed by several datasets like QuAC (Choi et al., 2018), CoQA (Reddy et al., 2019), and TREC CAsT (Dalton et al., 2019, 2020, 2021; Owoicho et al., 2022). A common strategy is concatenating previous questions with the current one for passage retrieval, yet this may be suboptimal for methods like BM25, which favour concise queries. Several studies (Aliannejadi et al., 2020; Mele et al., 2020; Ríssola et al., 2019; Sekulić et al., 2020) have offered alternatives to improve retrieval performance. Mele et al. (2020) proposed detecting topic shifts through heuristics and dependency parsing, while Aliannejadi et al. (2020) and Ríssola et al. (2019) utilised BERT models to predict historical questions enhancing the current one's retrieval. Sekulić et al. (2020) furthered this by predicting related past questions and reformulating the current one using ALBERT. In contrast, our work focuses on using follow-up question identification to improve conversational question rewriting.

**Follow-up Question Identification:** This task has seen diverse methods, ranging from rule-based solutions (Bertomeu et al., 2006; Kirschner and Bernardi, 2007) to statistical machine learning models like Logistic Regression (Kirschner and Bernardi, 2009). Kundu et al. (2020) proposed a standout three-way attentive pooling network, identifying follow-up questions related to the conversation history and the associated answer passage, outperforming rule-based methods, logistic regression models, and neural network-based models such as

BiLSTM, CNN, and BERT (Devlin et al., 2019). In our study, we utilise both the three-way attentive pooling network and BERT as baselines, given their proven effectiveness.

**Conversational Question Rewriting:** This task involves transforming conversational queries into context-independent queries suitable for information retrieval (IR) systems (Mele et al., 2021). Approaches range from sequence-to-sequence models like LSTM and GRU (Ren et al., 2018), to rule and self-supervised learning methods fine-tuning GPT-2 (Yu et al., 2020), and Transformer++ trained on the CANARD dataset (Vakulenko et al., 2021a). Vakulenko et al. (2021b) compared various methods on the CAsT 2019 and 2020 datasets, emphasising the effectiveness of Transformer++. Recently, Lin et al. (2020) presented a T5 model for conversational question rewriting. T5 outperformed neural network-based models such as LSTM, GPT-2, BERT, and UniLM on the CANARD and CAsT 2019 datasets. In this paper, we propose a T5 model for conversational question rewriting due to its overall effectiveness.

**MTL in Conversational QA:** MTL has been applied in Conversational QA, primarily focussing on answer span prediction and auxiliary tasks (Kongyoung et al., 2020; Qu et al., 2019; Xu et al., 2019). Some studies have combined classification and text generation tasks to improve passage ranking and answer generation (Ide and Kawahara, 2021; Kongyoung et al., 2022; Lee et al., 2022; Jiang et al., 2022), often leveraging models such as T5 (Raffel et al., 2020). Similarly, Jiang et al. (2021) combined the classification of relation types and the generation of sentences to express these relation types. Moreover, Wang et al. (2020) aimed to improve the response selection in multi-party conversations using an auxiliary task, namely a topic prediction task, which classifies whether a follow-up question is relevant to the user. Our work also embraces the MTL paradigm, but unlike prior studies prioritising answer generation, response selection, and passage ranking, we focus on improving retrieval effectiveness by rewriting the user's question using a classification task.

**Other Approaches Addressing Ambiguity in Conversational Questions:** Shao et al. (2022) addressed the ambiguity of the conversational question by asking clarifying questions to the user. Similarly, Li et al. (2023) explored enhancing dialogue generation with conversational concept flows, using a conversation-aware knowledge graph and a novel relation-aware graph encoder. In contrast to these studies, our work focuses on addressing the ambiguity of the conversational question by reformulating the user's original question and classifying whether the conversation is still focused on the same topic to enhance the accuracy of the information retrieval system.

# 6 Conclusions

We proposed a method for Conversational QA, which learns to predict the follow-up question and rewrites the conversational question simultaneously. Our proposed MTL method makes use of text generation models including BART and T5 by generating the first token for a classification task and the follow-up tokens for a conversational question rewriting task. For the follow-up question identification task, our experiments on the LIF dataset showed that our proposed MTL BART model has the best effectiveness, yielding statistically significant improvements over the baselines. For the question rewriting task, our proposed MTL T5 model performed best both in terms of first stage retrieval and re-ranking. Furthermore, we showed that the BART-based model can be employed for the follow-up question identification and the conversational question rewriting tasks in both discriminative+generative and generative MTL setups.

**Limitations and Future Work**

While our work has demonstrated promising results in leveraging multi-task learning for conversational question rewriting, several limitations must be acknowledged. Firstly, our model does not currently incorporate user interaction or feedback. This means it may not always accurately capture user intent or adjust to evolving conversational dynamics. In addition, without user feedback, the model lacks a mechanism for continuous learning and improvement from real-world applications. Finally, the model does not currently ask clarifying questions (Owoicho et al., 2022; Aliannejadi et al., 2021) when faced with ambiguous queries. This could limit its effectiveness in certain complex or unclear conversational scenarios. In future work, we plan to address these limitations by integrating user interaction and feedback, and by developing a mechanism for the model to ask clarifying questions when necessary.

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

# A Appendix

Our code and data are publicly available at the following URL: `https://github.com/terrierteam/mtl_gen_class`.

## A.1 A Discriminative+Generative MTL Model

In addition to the implementation of the generative MTL model, we also incorporate a discriminative+generative MTL model to capture the relationship between the question $q_c/q_m$ and the contextual information in the conversation history, which includes the historical question(s) $q_1, q_2, \ldots q_k$ and the historical answer(s) $a_1, a_2, \ldots a_k$, as illustrated in Figure 2(a). The input sequence of the discriminative+generative MTL model is the same as the one shown in Equation (1). During the fine-tuning process, the LM head of the model is refined to generate the target tokens, with a token sequence length denoted as $n$ in Equation (2). In contrast to

the generative MTL model, this approach employs the tokens $w_1, ..., w_n$ to encompass the entire output sequence for the target query. Simultaneously, the CLS head (see details in Section 2.3) of the model undergoes fine-tuning to predict either 1 or 0, signifying whether the candidate question is a valid follow-up to the previous question or not.

We compare our generative BART model to the discriminative+generative BART variant in Section 4.1 and Section 4.2, to conclude on their performance in both the follow-up question identification and conversational question rewriting tasks.

| Model Training Details | |
|---|---|
| Mathematical setting | Section 2.3, Appendix A.1 |
| Source code | https://github.com/terrierteam/mtl_gen_class |
| Computing infrastructure | NVIDIA RTX A6000 GPU |
| Training time | 4h |
| Inference time | 500ms |
| Batch size | 24 |
| Number of parameters in each model | 220M (MTL T5) 140M (MTL BART) |
| Evaluation metrics | Appendix A.2 |
| **Hyper-parameter Experiments** | |
| The exact number of training and evaluation runs | Appendix A.2 |
| Bounds for each hyper-parameter | Number of epochs: 1–5 |
| Hyperparameter configurations for best-performing models | Appendix A.2 |
| Number of hyperparameter search trials | 1 |
| The method of choosing hyper-parameter values | Highest question rewriting effectiveness (ROUGE-1) on validation set |
| **Dataset** | |
| Dataset languages | English |
| Number of examples in datasets | Table 1 |
| Explanation of any data that were excluded, and all pre-processing steps | Section 3.2 |
| A zip file containing data or link to a downloadable version of the data | https://github.com/terrierteam/mtl_gen_class |

Table 4: Summary of Reproducibility Criteria.

## A.2 Hyperparameter Settings

We implement the BERT, GPT-2, BART, and T5 models using the following PyTorch models from HuggingFace (Wolf et al., 2020), namely `bert-base`, `facebook/bart-base`, and

`ram srigouthamg/t5_paraphraser`:[3] These models are configured as follows:[4] the maximum sequence length is set to 512, the number of training epochs is set to 5, the batch size is set to 24, and we use Adam optimiser with a learning rate of 0.00005.

---

[3]Initial experiments found this T5 model more effective than the original `t5-base` across a number of tasks.
[4]Settings follow https://github.com/gmihaila/ml_things/

## A.3 Evaluation Metrics

Since the follow-up question identification is a binary classification task, we evaluate performances using classical classification metrics, namely precision, recall, F1 and Macro-F1. Indeed, following (Kundu et al., 2020), reporting Macro-F1 enables accuracy on topic shift detection to be measured, while F1 focuses solely on follow-up identification as the positive class. We use McNemar's test to measure statistically significant differences between the models' classification performances. For the evaluation of the conversational question rewriting performance, we adopt the ROUGE recall calculated for unigrams (ROUGE-1 recall) and BLEU metrics, following (Vakulenko et al., 2021a; Lin et al., 2020; Elgohary et al., 2019). As for the passage retrieval evaluation, we use Mean Average Precision (MAP), Mean Reciprocal Rank (MRR), Normalized Discounted Cumulative Gain (NDCG) and Recall@1000 as metrics. For each query, the top 1000 documents are considered. We use the paired t-test for testing significant differences.

**Passage Retrieval Pipeline:** We use the PyTerrier (Macdonald and Tonellotto, 2020) platform for indexing and retrieving passages. For passage ranking, we incorporate BM25 with the monoT5 (Pradeep et al., 2021) re-ranker.

## A.4 Reproducibility Criteria

Table 4 summarises the reproducibility criteria questions for this paper.