# OpenReview forum: "Multi-Task Learning of Query Generation and Classification for Generative Conversational Question Rewriting"
_EMNLP/2023/Conference — EMNLP 2023 Findings_

### Official Review · Reviewer_Y11v · 2023-08-04

**Soundness:** 3

**Excitement:**

3: Ambivalent: It has merits (e.g., it reports state-of-the-art results, the idea is nice), but there are key weaknesses (e.g., it describes incremental work), and it can significantly benefit from another round of revision. However, I won't object to accepting it if my co-reviewers champion it.

**Missing References:**

The following reference/s are missing from Conversational QA related work.
* Raviteja Anantha,Svitlana Vakulenko,Zhucheng Tu,Shayne Longpre,Stephen Pulman, and Srinivas Chappidi. 2021. Open-domain question answering goes conversational via question rewriting. In Proceedings of the 19th Conference of the North American Chapter of the Association for Computational Linguistics: Human Language Technologies (NAACL-HLT’21). 520–534. https://aclanthology.org/2021.naacl-main.44.

**Paper Topic And Main Contributions:**

Authors propose a Multi-task Learning (MTL) for the problem of conversational QA where the model learns both to identify a context-dependent follow up question as well to rewrite one into a self-contained form by resolving references. Authors perform experiments using BART and T5 models, on a OR-QuAC test set authors show improvement over single-task learning (STL) for passage retrieval.

**Questions For The Authors:**

* Did you try experimenting w/ bigger models and try different retrieval and reranking methods to see how that affects the magnitude of the potential improvements from your MTL approach?

**Reasons To Accept:**

* It's an interesting find that combining follow up identification w/ question rewriting is improving the passage retrieval.

**Reasons To Reject:**

* While the results show stat-sig improvements, I still intuitively feel the improvement in results across a wider experiment setting (e.g. different retrieval and reranking methods, more QR models, bigger models etc) might not be significant.

**Reproducibility:**

3: Could reproduce the results with some difficulty. The settings of parameters are underspecified or subjectively determined; the training/evaluation data are not widely available.

**Reviewer Confidence:**

5: Positive that my evaluation is correct. I read the paper very carefully and I am very familiar with related work.

---

> ### Author Rebuttal · Authors · 2023-08-28
>
> *While the results show stat-sig improvements, I still intuitively feel the improvement in results across a wider experiment setting (e.g. different retrieval and reranking methods, more QR models, bigger models etc) might not be significant.*
>
> We appreciate the reviewer's suggestion to consider a broader experiment setting. However, it is worth mentioning that the models we used for evaluation are state-of-the-art and have already shown generalisability across various tasks and datasets. In the paper, we employed the OR-QuAC dataset, using a single but highly competitive retrieval and reranking method (BM25 + monoT5).
> Due to space constraints, we were unable to report additional results demonstrating the generalisability of our approach. In particular, in addition to the BM25 + monoT5 (extracted from Table 2 in the paper), the table below reports performances using a BERT reranker (Nogueira et al., 2020c) over the DPH weighting model. The results here illustrate that our approach generalises over the choice of retrieval pipeline.
>
> | Retriver+Reranker | Models |   MAP   |   MRR  |  NDCG  |
> |-------------------|:------:|:-------:|:------:|:------:|
> | BM25+monoT5       | STL T5 | 0.2783  | 0.2957 | 0.3430 |
> |                   | MTL T5 | 0.3106  | 0.3302 | 0.3764 |
> | DPH+BERT          | STL T5 | 0.2771  | 0.2949 | 0.3414 |
> |                   | MTL T5 | 0.3081  | 0.3286 | 0.3735 |
>
>
> That said, we understand the value in diversifying the experiment setting, including different retrieval and reranking methods as well as other datasets. We believe that such extensions, while important, do not diminish the current contributions of our paper. We will consider this in our future work to provide a more comprehensive evaluation.

---

### Official Review · Reviewer_yEsQ · 2023-08-05

**Soundness:** 4

**Excitement:**

3: Ambivalent: It has merits (e.g., it reports state-of-the-art results, the idea is nice), but there are key weaknesses (e.g., it describes incremental work), and it can significantly benefit from another round of revision. However, I won't object to accepting it if my co-reviewers champion it.

**Paper Topic And Main Contributions:**

This study explores a multi-task learning approach for multi-topic question rewriting, which includes an additional task to identify follow-up questions. According to the authors, this is the first proposal of such a combination. A seq2seq model is used to predict the rewriting and the classification, and two alternative implementations are discussed. The authors conduct experiments on two datasets, one for each task, and report superior performance against recent baselines.

**Questions For The Authors:**

Question A: I am curious about a more general setting: identify related turns in the conversation because I think a topic could be recalled. For example, for a conversation (q1, a1)(q2, a2)(q3, a3),   (q1, a1)(q3, a3) share the same topic while (q2, a2) is about another topic. Then would it be beneficial to find the recalled topic instead of just classifying whether it is a follow-up question.

Question B: Some words say (e.g., Sec 3.2) LIF and CANARD are used for evaluation, while some others say (e.g., Abstract, Introduction, and Sec 4.2) LIF and OR-QuAC are used. Which one is correct?

**Reasons To Accept:**

1. The task discussed is well-defined and well-motivated.

2. The proposed approach is simple and proven to be effective, according to comprehensive experiments.

**Reasons To Reject:**

1. There are closely related works that need to be discussed. This paper focuses on the multi-topic scenario. But the topic is not the only dynamic state during a conversation, such as concepts and indents (e.g., [1][2]). There are works on tracing the dynamics of these states. A broader discussion is expected.

2. ~~The authors do not follow the writing instructions from EMNLP. The captions of the table are misplaced.~~

[1] Siheng Li, Wangjie Jiang, Pengda Si, Cheng Yang, Qiu Yao, Jinchao Zhang, Jie Zhou, and Yujiu Yang. 2023. Enhancing Dialogue Generation with Conversational Concept Flows.
[2] Weishi Wang, Steven C.H. Hoi, and Shafiq Joty. 2020. Response Selection for Multi-Party Conversations with Dynamic Topic Tracking.

**Reproducibility:**

5: Could easily reproduce the results.

**Reviewer Confidence:**

2: Willing to defend my evaluation, but it is fairly likely that I missed some details, didn't understand some central points, or can't be sure about the novelty of the work.

---

> ### Author Rebuttal · Authors · 2023-08-28
>
> *Reasons to Reject 1: There are closely related works that need to be discussed.*
>
> Thank you for the constructive feedback. We acknowledge the importance of a broader discussion of related works that focus on other dynamic states during conversations, such as concepts and intents. In our revised manuscript, we will consider expanding our discussion to cover these works and clarify how our paper complements or differs from them.
> For example, while our paper uses Multi-Task Learning to focus on follow-up question identification and conversational question rewriting, Wang et al., 2020 aims to improve response selection in multi-party conversations by considering multiple topics. Similarly, Li et al., 2023, explores enhancing dialogue generation with conversational concept flows, using a conversation-aware knowledge graph and a novel relation-aware graph encoder.
>
>
> While these works provide valuable insights into other aspects of conversational dynamics, they do not serve as direct baselines for our study. This is because our focus is on follow-up question identification and conversational question rewriting, objectives that are distinct from the aims of the aforementioned papers. Consequently, while a broader discussion of these related works will provide valuable context (and we can add these as wider context in the next version of the paper, using the customary extra page), their absence in our study does not represent missing baselines, and hence do not detract from the validity or integrity of our results.
>
>
>
>
> *Reasons to Reject 2: The authors do not follow the writing instructions from EMNLP. The captions of the table are misplaced.*
>
>
> Thanks for highlighting this. We'll make this 2-second fix in the next version of paper, but, with all due respect, this is not a reason for rejection.
>
>
> *Questions For The Authors 1:*
>
> *Question A: I am curious about a more general setting: identify related turns in the conversation because I think a topic could be recalled. For example, for a conversation (q1, a1)(q2, a2)(q3, a3), (q1, a1)(q3, a3) share the same topic while (q2, a2) is about another topic. Then would it be beneficial to find the recalled topic instead of just classifying whether it is a follow-up question.*
>
>
> Thank you for raising this interesting point. Identifying related turns in a conversation based on recurring topics could certainly enhance the system's ability to generate more relevant and accurate responses. While our current work focuses on follow-up question identification and question rewriting, incorporating topic recall as an additional dynamic feature in the model could be beneficial. This approach could further refine the system's understanding of the conversation and thereby improve its responses. We appreciate the suggestion and will consider exploring this in future work.
>
> *Question B: Some words say (e.g., Sec 3.2) LIF and CANARD are used for evaluation, while some others say (e.g., Abstract, Introduction, and Sec 4.2) LIF and OR-QuAC are used. Which one is correct?*
>
> Thank you for catching that inconsistency. To clarify, we experiment using the LIF dataset and the CANARD dataset, which are a recent adaptation of the well-known QuAC Conversational QA dataset. For the training and development sets, we integrate LIF and CANARD by picking only the candidate questions from the LIF dataset that exist in the CANARD dataset. We used the OR-QuAC dataset for evaluation, which is an aggregation of the QuAC, CANARD, and Wikipedia datasets. This allows us to assess our model's performance across conversational question answering and passage retrieval tasks, as detailed in Appendix A.2.

---

### Official Review · Reviewer_PJ74 · 2023-08-11

**Soundness:** 3

**Excitement:**

3: Ambivalent: It has merits (e.g., it reports state-of-the-art results, the idea is nice), but there are key weaknesses (e.g., it describes incremental work), and it can significantly benefit from another round of revision. However, I won't object to accepting it if my co-reviewers champion it.

**Missing References:**

Jiang, Feng, Yaxin Fan, Xiaomin Chu, Peifeng Li and Qiaoming Zhu. “Not Just Classification: Recognizing Implicit Discourse Relation on Joint Modeling of Classification and Generation.” Conference on Empirical Methods in Natural Language Processing (2021).
Shao, Taihua, Fei Cai, Wanyu Chen and Honghui Chen. “Self-supervised clarification question generation for ambiguous multi-turn conversation.” Inf. Sci. 587 (2021): 626-641.

**Paper Topic And Main Contributions:**

This paper uses BART and T5 to learn simultaneously a generation task (question rewriting) and a classification tasks (whether a question is a follow-up to the previous one). The presented approach shows good performance compared to the single-task learning baselines on the LIF dataset (Kundu et al., 2020) and OR-QuAC.

**Reasons To Accept:**

1. The topic of question rewriting/identification is well-introduced, although it would be great to include additional pointers to the similar task of clarification question identification and generation. There are some recent works on this topic (Shao et al., 2022) and even a shared task with the special track for CR identification organized at NeurIPS last year (Kiseleva et al., 2022).

2. The authors perform a good comparison of the models using significance testing. The results look robust and show an improvement over the selected baselines. The computational budget and training details are reported.

**Reasons To Reject:**

1. The idea of joint text generation and classification with T5 is quite straightforward and lacks novelty (see e.g., (Jiang et al., 2021) for similar experiments with implicit discourse relation classification/generation). Besides, all experiments are only for English.

2. The experimental results section is a bit cluttered and difficult to read. Also, although the authors mention that their code and data are publicly available there are only empty README files in the corresponding repository in anonymous.4open.science (solved).

3. The authors use ROUGE-1 and BLEU for the question rewriting evaluation. However, the shortcomings of these metrics are quite known (see e.g., the following ACL publications:(Post, 2018), (Sulem et al., 2018), (Mathur et al., 2020) and there exist some alternatives like BERTScore or BLEURT. I think they should be also included to make the study more robust/comprehensive.

**Reproducibility:**

4: Could mostly reproduce the results, but there may be some variation because of sample variance or minor variations in their interpretation of the protocol or method.

**Reviewer Confidence:**

4: Quite sure. I tried to check the important points carefully. It's unlikely, though conceivable, that I missed something that should affect my ratings.

**Typos Grammar Style And Presentation Improvements:**

In section 4 (Experimental Results) you refer to Test-I, II, III without explaining what is the difference between these tests.

---

> ### Author Rebuttal · Authors · 2023-08-28
>
> *Reasons to Reject 1: There are closely related works that need to be discussed. This paper focuses on the multi-topic scenario. But the topic is not the only dynamic state during a conversation, such as concepts and indents (e.g., [1][2]). There are works on tracing the dynamics of these states. A broader discussion is expected.*
>
> Thank you for pointing out the work by Jiang et al., 2021 [1]. While it's true that they also use Multi-Task Learning involving both classification and text generation, their focus is on Implicit Discourse Relation Recognition (IDRR). Their model is trained to detect relation types and generate sentences that express these relations. In contrast, our work is focused on improving conversational search systems. We use models based on BART and T5 to rewrite conversational questions and identify follow-up questions simultaneously. These are substantially different applications and objectives.
>
> The work of Shao et.al., 2021 [2] focused on addressing the ambiguity of the conversational question by *asking* clarifying questions to the user. In contrast, our work is focused on addressing the ambiguity of the conversational question by reformulating the user's original question to enhance the accuracy of the retrieval information system. Hence, this addresses a different user task.
>
> We are happy to cite [1] & [2]. in our final version of the paper, but their omission from this stage do not detract from the merits of our paper.
>
>
>
> *Reasons to Reject 2: The experimental results section is a bit cluttered and difficult to read. Empty readme in anonymous repository.*
>
> We appreciate the feedback on the organisation of the experimental results section. We plan to improve the readability and clarity of this section in the revised version using the extra page that is customarily made available. To make this feedback more actionable, clarifying your review with specific problems would be appreciated.
>
> As for the code and data repository, we apologise for the delay. We have now improved our anonymous github repo to facilitate a more thorough understanding and replication of our proposed method. In particular, the training codes and training instructions have been provided.
>
>
>
> *Reasons to Reject 3: Shortcomings of ROUGE-1 and BLEU*
>
> We appreciate your concern about the shortcomings of ROUGE-1 and BLEU. In our paper, we chose to use ROUGE-1 and BLEU because they are the most widely used metrics for question rewriting evaluation. They have been shown to be effective in many previous studies, and they are relatively easy to compute.
> We think the suitability of more advanced measures (such as BERTScore and BLEURT) would require a separate study focussed on evaluation. In contrast, the scope of this paper is upon models rather than evaluation methods. In addition, we measure the downstream retrieval effectiveness of the reformulated queries (e.g. nDCG, MAP), which verifies the conclusions we observe from ROUGE-1 (Vakulenko et al., 2021a).
>
>
>
> *Questions For The Authors:In section 4 (Experimental Results) you refer to Test-I, II, III without explaining what is the difference between these tests.*
>
> In the revised version, we will provide further clarification on the differences between the Test-I, II, and III sets in the LIF dataset. In all three, the valid follow-up questions (label = 1) are constructed from the “should ask follow-up question” instances in the QuAC dataset. On the other hand for Test-II, questions with high similarity to the current passage are sampled from other conversations; for Test-III, the invalid follow-up questions are sampled from non-follow-up questions from the same conversation in QuAC; Test-I combines invalid instances from  Test-II & Test-III. The training and the development sets follow the same procedure as Test-I.In particular, the training and development sets contain 126k and 5.8k instances each; Test-I, II & III all have 1940 positive instances, and respectively 745, 3307 and 4052 negative instances.

---

### Meta-Review · Area_Chair_sGoq · 2023-09-13

**Recommendation:** 2

**Metareview:**

This work considers the conversational search setting. Primarily, the work proposes an MTL approach which simultaneous considers topic continuity and question reformulation (instead of independently).

**Pros**: Reviewers tend to agree the task and motivations/hypotheses are well-defined with generally convincing experiments (among those conducted). Some reviewers also comment on interesting takeaways from the work.

**Cons**: After rebuttal, there is a shared sentiment among a few reviewers that the scope of the experiments is limited (1 language, missing perturbations among important variables like larger models and retrieval system, etc.). Authors do provide some new experiments, but they do not address all concerns. For one reviewer, this issue is compounded by substantial similarity to another MTL paper looking at classification/generation of implicit discourse relations. While this work is arguably very similar, and other MTL approaches abound in the current literature, the authors suggest the importance of the conversational setting in the current work. Besides these concerns on soundness and scope, reviewers are generally ambivalent - there is no champion.

---

### Decision · Program_Chairs · 2023-10-07

**Decision:**

Accept-Findings

**Comment:**

This work considers the conversational search setting. Primarily, the work proposes an MTL approach which simultaneous considers topic continuity and question reformulation (instead of independently).

**Pros**: Reviewers tend to agree the task and motivations/hypotheses are well-defined with generally convincing experiments (among those conducted). Some reviewers also comment on interesting takeaways from the work.

**Cons**: After rebuttal, there is a shared sentiment among a few reviewers that the scope of the experiments is limited (1 language, missing perturbations among important variables like larger models and retrieval system, etc.). Authors do provide some new experiments, but they do not address all concerns. For one reviewer, this issue is compounded by substantial similarity to another MTL paper looking at classification/generation of implicit discourse relations. While this work is arguably very similar, and other MTL approaches abound in the current literature, the authors suggest the importance of the conversational setting in the current work. Besides these concerns on soundness and scope, reviewers are generally ambivalent - there is no champion.